# Modified combined short and long axis method versus oblique axis method in adult patients undergoing right internal jugular vein cannulation: A randomized controlled non-inferiority study

Jia-Xi Tang[1,2], Ling Wang[3], Ju Ouyang[4], Xixi Tang[2], Mengxiao Liu[2], Hongliang Liu[2]*, Fang Xu[1]*

1 Department of Critical Care Medicine, The First Affiliated Hospital of Chongqing Medical University, Chongqing, China, 2 Department of Anesthesiology, Chongqing University Cancer Hospital, Chongqing, China, 3 Department of Phase I Clinical Trial Ward, Chongqing University Cancer Hospital, Chongqing, China, 4 Department of Oncology, Hechuan District Hospital of Integrated Chinese and Western Medicine, Chongqing, China

☯ These authors contributed equally to this work.

* liuhl75@163.com (HL); xufang828@126.com, fangxu@hospital.cqmu.edu.cn (FX)

**Data Availability Statement:** All metadata files are available from the DRYAD database (URL: https://

## Abstract

### Background

Modified combined short and long axis method (MCSL) can replace oblique axis in-plane method (OA-IP) for internal jugular vein cannulation (IJVC). This randomized, non-inferiority study estimated the efficacy of MCSL compared with OA-IP in right IJVC.

### Methods

Patients (18–75 yr. old) undergoing right IJVC under local anesthesia were randomly assigned to MCSL or OA-IP group. The primary outcome is the event of first needle pass without posterior vessel wall puncture (PVWP). Secondary outcomes included needle attempts, success rate, puncture and cannulation time, needle visualization, probe placement difficulty and complications.

### Results

Among 190 randomized patients, 187 were involved in the analysis. The first needle pass without PVWP was 85(89.47%) in the MCSL and 81 (85.26%) in the OA-IP (p = 0.382), with a mean rate difference of 4.2% (95% confidence interval: -5.2–13.6), which confirmed the non-inferiority with the margin of -8%. MCSL group exhibited shorter procedure time and lower complications than OA-IP group. No significant differences were discovered between groups in needle attempts, success rate, incidence of probe placement difficulty and needle visualization.

datadryad.org/stash/share/
AKa3Ii55O3CidSErLGro7H-_Js2cl_i3ntlnOlPQ31c).

**Funding:** This research was funded by Key R&D project of Chongqing Science and Technology Bureau grant to HLL(Cstc2020jscx-dxwtBX0010, http://kjj.cq.gov.cn/) and Key Laboratory Open Fund Project of Chongqing University Cancer Hospital grant to JXT (cquchkfjj006, https://www.cqch.cn/). The funders had no role in study design, data collection and analysis, decision to publish, or preparation of the manuscript.

**Competing interests:** The authors have declared that no competing interests exist.

## Conclusions

MCSL is non-inferior to OA-IP in first needle pass without PVWP in adults who underwent elective right IJVC and associate with less complications and shorter operating time.

## Clinical trial registration

ChiCTR, ChiCTR2100046899.

## Introduction

Several studies support the subclavian site as the optimal choice for central vein catheterization (CVC), minimizing catheter-related infections [1–4]. In the high-risk ICU environment, where patients are typically more critically ill and require prolonged catheterization, subclavian access may be the preferred option [5–7]. In a sterile operating room environment, the internal jugular vein (IJV) is also a common location for establishing venous access due to its low risk of mechanical complications [1, 2]. Ultrasound guided cannulation has proven to be safer and better than landmark technique regardless the cannulation site [8–11]. Hence, it has been recommended as a routine procedure by many guidelines [2, 12–14].

Various ultrasound-guided techniques have been described for internal jugular vein catheterization [15–21]. The oblique-axis in-plane method (OA-IP) combines the benefits of short-axis view and in-plane method, under which, the adjacent relationship between common carotid artery (CCA) and IJV is clear, and the entire needle body is visible, therefore it is recommended to conduct internal jugular vein cannulation (IJVC) by many scholars [15, 22–25]. When we use the oblique axis plane method, we need to place the ultrasound probe at a 45-degree angle with the IJV and insert the needle in the plane.When using a single operator method to perform IJVC, the hands may be crossed, resulting in uncoordinated puncture procedures. This incoordination is more pronounced when a right-handed operator performs left internal jugular vein cannulation. In addition, the puncture point of oblique axis method is also close to some important structures , such as thyroid, external jugular vein and brachial plexus nerve. These may increase the risk of mechanical complications.

In combination with previous studies, we proposed modified combined short and long axis method (MCSL) [16, 17, 20, 26]. Similar to the OA-IP method, the MCSL method can accurately locate the puncture needle point and visualize the entire needle body throughout the procedure of IJVC. However, the comparative study between the MCSL and OA-IP is blank. We hypothesized that MCSL would not be inferior to OA-IP regard to first needle pass without posterior vessel wall puncture (PVWP) during IJVC. Based on panel discussion, we predefined a non-inferiority margin of -8% for the difference in the primary outcome between groups. To test our hypothesis, we designed a prospective clinical trial to evaluate the safety and efficacy of MCSL versus OA-IP for right IJVC in adult patients.

## Methods

### Study design and participants

This prospective randomized, evaluator-blind, parallel-controlled, non-inferiority trial was approved by the institutional ethics committee on February 4, 2021(CZLS2021042-A), and was registered at ChiCTR.org.cn (ChiCTR2100046899) May 30, 2021. The protocol of our study has been published in advance [27] (S1 Text). It was managed according to the Helsinki

declaration, Consolidated Standards of Reporting Trials statement (CONSORT) and Extension of the CONSORT 2010 Statement for Non-inferiority and Equivalence Randomized Trials [28, 29] (S1 Table). All written informed consents were provided by subjects before they were enrolled in our trial.

The patients visited from June 18, 2021 to November 2, 2021 and aged from18 to 75 years old scheduled to undergo elective surgery requiring right IJVC in a tertiary hospital (Chongqing University Cancer Hospital, Chongqing, China) were screened for eligibility in the study the day before IJVC (Fig 1). Patients met any following criteria (Table 1) were excluded.

To ensure equal distribution of participants across intervention groups during the study, a block of 4 randomization approach with a 1:1 allocation ratio was employed. The randomization process was conducted using a computer-generated random number sequence, which was managed by the study biostatistician. Patients undergoing right IJVC were randomly assigned to receive either MCSL or OA-IP. Follow-up was conducted for a duration of 3 days.

Opaque envelopes were used to maintain allocation concealment. Considering the unique aspects of block randomization, our randomization plan was developed by statistical experts. The detailed randomization method was not disclosed to the operators and outcome assessors. The random allocation plan was securely maintained by statistical experts, and unblinding occurred only after the completion of experimental and statistical analyses. Because of the specificity of the puncture procedure, patients and investigators were not blinded to group assignment. Data monitors and statistical analysts were blinded in this study. Prior to the completion of data analysis, only the enrollment numbers of subjects were recorded, and the group names were replaced with A or B.

## Intervention

In order to minimize the bias, all IJVC processes were completed by the same attending doctor who had completed ultrasound-guided IJVC for more than 500 cases. All IJVC procedures were performed under local anesthesia and using the Seldinger method, with a 6.35cm, 18-gauge needle (BIOSENSORS INTERNATIONAL™) attached by a 5 ml syringe. In the supine position, patients tilted their head to the leftward 30˚, with a thin pillow under their shoulders to extend the neck. All patients underwent vital sign monitoring during the IJVC procedure. The same ultrasound equipment (UMT-500; Mindray, Shenzhen, China) with a high-frequency linear (8 - 13Hz) transducer of footprint size 47 mm was used for ultrasound guidance.

In the MCSL group, IJVC procedures were conducted as described in our previous report [17]. Before IJVC, the ultrasound probe was prepared using two silk sutures parallelly tied on the ultrasound probe (interval: 10 mm). Firstly, the ultrasound probe was placed longitudinally in the supraclavicular fossa to determine the puncture plane, which was on the side of the head above the red line. Then adjusted the ultrasound probe so that the double shadows were located above the IJV, and inserted the needle between the two lines on the probe at a 30˚ angle to the skin using the short-axis out-of-plane technology. When the tip of needle was shown as a white dot on the ultrasound screen, stopped the needle. Finally, rotated the ultrasound probe 90˚, then punctured IJV's anterior wall using the long axis in-plane approach (Fig 2A–2C). Patients in the OA-IP group used the lateral oblique approach was described in the previous study [25]. Firstly, the ultrasound probe was placed parallelly to the clavicle to obtain a short-axis view of the CCA and IJV, and then the probe was rotated 45˚ clockwise to obtain CCA and IJV's oblique-axis view. Finally, inserted the needle from lateral to medial using the in-plane approach (Fig 2D). When the guide wire was placed into the vessel, removed the syringe and identified the guide wire in the IJV under the ultrasound short-axis view.

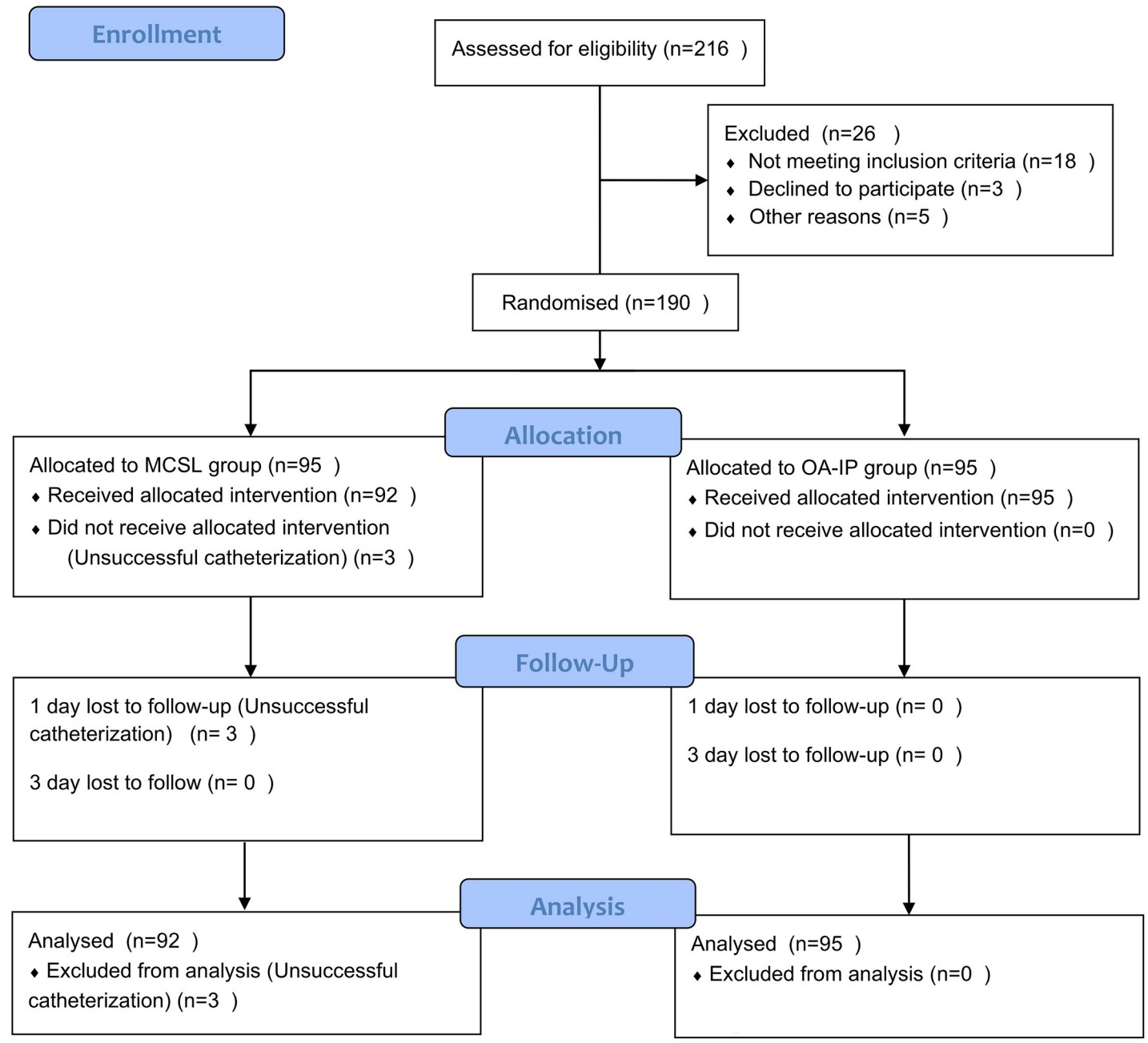

**Fig 1. The CONSORT study flow chart.** MCSL, modified combined short and long axis method; OA-IP, Oblique axis in-plane method.

Following the fixation of catheter, chest radiographs and ultrasonic examination were performed to check mechanical complications.

Before skin preparation, pre-ultrasonography was performed to verify patency of the right IJV and its sonoanatomy parameters (the diameter of IJV, the area of IJV (trace method), depth of IJV and the overlap between IJV and CCA under the short axis view) were measured. The following clinical variables were also recorded: age, sex, height, weight, neck length (from clavicle to mandibular border), fasting time, preoperative infusion volume, bowel preparation situation and the type of catheter (single (5 French) or double lumen (7 French) catheters).

**Table 1. Patient exclusion criteria.**

| |
|---|
| 1. Failure to provide consent |
| 2. Abnormal blood coagulation function (INR > 1.5, platelet count < 50.000) |
| 3. History of previous surgical intervention or radiotherapy near the puncture site |
| 4. Huge masses or lymph nodes present in the right neck |
| 5. Infection signs, neck scar, subcutaneous hematoma or emphysema at the puncture site |
| 6. Cervical trauma with present neck immobilization |
| 7. History of right IJV catheterization within the past 1 month |
| 8. Presence of superior vena cava syndrome or right IJV thrombosis |
| 9. patients with anatomical variations and no right IJV |
| 10. ASA physical status greater than 3 |
| 11. Agitated or uncooperative patient |
| 12. Right chest surgery |
| IJV, internal jugular vein; INR, international normalized ratio |

## Study endpoints and sample size calculation

Successful first needle pass without PVWP was the primary outcome of our study. Another needle pass was defined as every needle withdrawal with subsequent redirect and advance, regardless of whether another skin puncture site was selected. Unsuccessful catheterization was identified as the following conditions, including probe placement difficulty (due to the patient's short neck, there was insufficient space to place the ultrasound probe for ultrasound-guided CVC) results in no space for puncture, multiple attempts (>3 ) or the occurrence of serious complications (artery puncture, hematoma, nerve damage, thyroid injury, pneumothorax, and hemothorax). According to the ultrasound image of the puncture video, the needle tip penetrated the posterior wall of the IJV and was considered PVWP [30]. Secondary outcomes were needle attempts, successful puncture rate, incidence of probe placement difficulty, guide-wire insertion time (the time from needle insertion to ultrasonic confirmation of guide wire in IJV), total cannulation time (from the beginning of skin puncture to completion of central venous catheter fixation) , needle visualization (binary variable, "yes" or "no," indicating real-time needle tracking), probe placement difficulty and incidence of mechanical complications (the diagnosis was confirmed by ultrasonography, other imaging examination and clinical evaluation).

Expected mechanical complications included: artery puncture (any pulsating blood return to the needle observed during the IJVC procedure or the puncture needle into the artery was observed through the puncture video), external jugular vein (EJV) puncture (the puncture needle into the EJV was observed through the puncture video), PVWP, hematoma (identified by ultrasound) , nerve damage (included vagus, spinal, and brachial plexus injury) , thyroid injury, pneumothorax and hemothorax (identified by chest radiograph).

Two evaluators who did not perform the IJVC procedures evaluated the aforementioned outcomes through puncture video. The two evaluators had 5 yrs. of experience in point-of-care ultrasonography and had performed over 300 vascular access procedures under ultrasound guidance. The two judges made their own judgments based on the video. Any disagreements between the two assessors were resolved through discussion. If no consensus was reached, another assessor would be consulted.

According to the unpublished data from our pilot research, the rates of first needle pass without PVWP were 95% and 90% for MCSL and OA-IP methods, respectively. Based on the preset non-inferiority margin of -8% in the primary outcome, we utilized PASS software

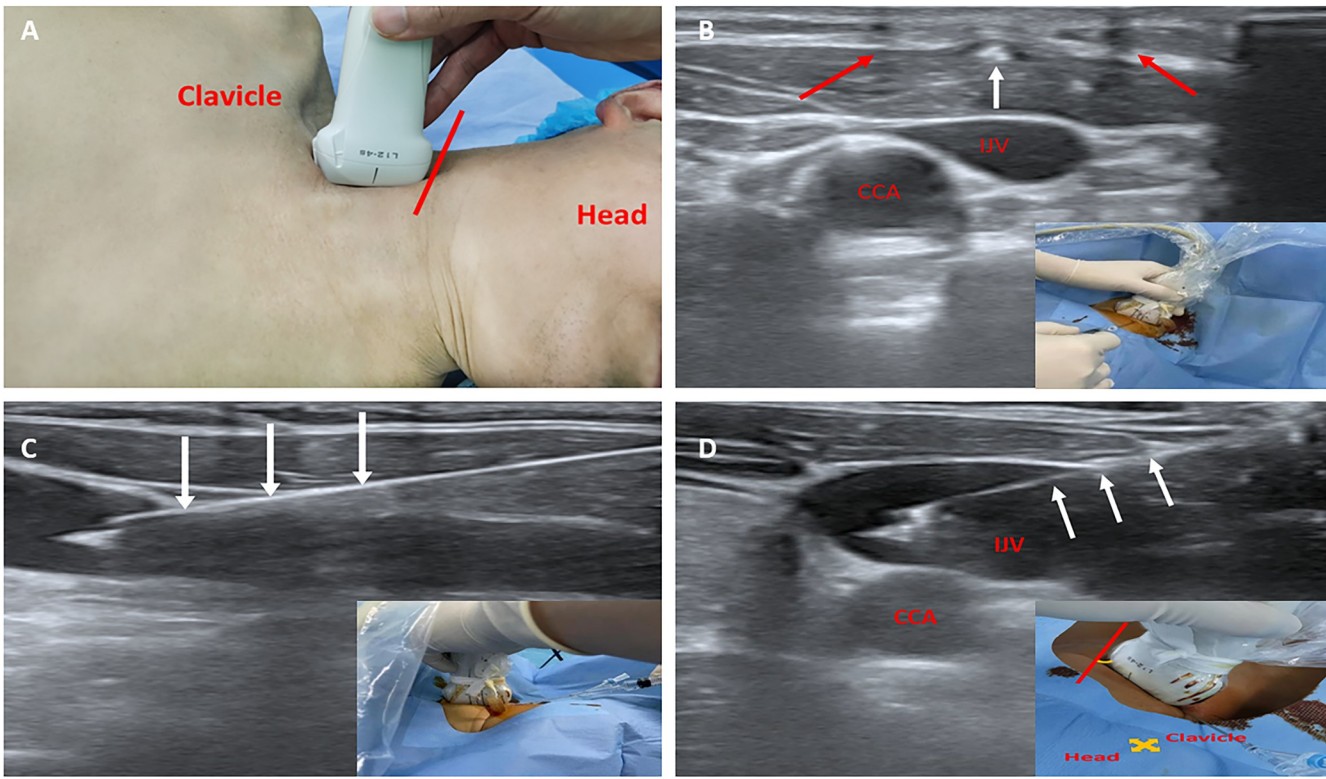

**Fig 2. MCSL and OA-IP for IJV catheterization.** All catheterization were performed from the skull to the caudal in the neck region. (a)(b)(c) MCSL approach. (d) OA-IP approach. (a) Step 1: Determine the puncture plane. (b) Step 2: Short-axis out-of-plane puncture. (c) Step 3: Long axis in-plane puncture. (d) Lateral Oblique-axis in-plane puncture. MCSL, Modified combined short and long axis method; OA-IP, Oblique axis in-plane method; IJV, internal jugular vein; CCA, common carotid artery; (red arrows), Ultrasound shadows; (white arrows), Needle.

(version 15) to estimate that 86 patients per group would supply 90% power at an α level of 0.025 to compare the MCSL approach with the OA-IP method. Ultimately, we recruited 95 patients in every group to take into account the 10% dropout.

## Statistical analysis

Data were analyzed by intention and per-protocol according to the data subset in which they were located. Continuous quantitative data were presented as median (IQR) or mean (SD). Qualitative data were expressed using percentage and number. T-test, Mann-Whitney U test, exact Fisher tests or $\chi^2$ test were used for comparison between groups, according to the type and characteristics of the variables. The receiver operating characteristic (ROC) curve was conducted to confirm the cut-off value of continuous variable. To analyze the factors predicting "First past success without pvwp," a binary logistic regression using the enter method was employed. Before including the independent variables, a multicollinearity test was performed to ensure that the variables being included in the model had a variance inflation factor (VIF) of less than 3. We assessed non-inferiority for the primary endpoint using bilateral 95% CI [29]. P-values < 0.05 were considered statistically significant. SPSS Statistics 25.0 software (IBM Corporation, USA) was applied for all statistical analysis, GraphPad Prism 8.0.2 was used for graphing.

# Results

## Patient characteristics

190 patients were randomly divided into either MCSL or control OA-IP group (Fig 1). There were no significant differences in baseline characteristics (age, sex, BMI, length of the neck, sonoanatomy parameters of IJV, ASA physical status, fasting time, preoperative infusion volume, bowel preparation situation and the type of catheter) between the two groups (Table 2).

## Primary outcome

For the primary outcome, the first needle pass without PVWP in the MCSL group was 85 (89.5%) and in the OA-IP group was 81 (85.3%), no significant difference were found (Table 3). The difference in rate (MCSL vs OA-IP) was 4.2% (95% CI: -5.5–13.6), and the lower limit of the 95% confidence interval did not fall below -8%, hence the non-inferiority was confirmed. The ROC curve analysis was performed to determine the best cut-off values of continuous variable for predicting the first needle pass without PVWP (S1 Fig and S2 Table). The predictors of first needle pass without PVWP were IJV longitudinal diameter > 5.9mm (OR 7.45, 95% CI: 2.14–25.89, P = 0.002) and CCA transverse diameter > 5.1mm (OR 6.81, 95% CI: 1.77–26.18, P = 0.005) (Fig 3). The positive predictive accuracy of the prediction model is 97.6%, the negative predictive accuracy is 33.3%, and the overall predictive accuracy is 89.5%. The prediction model was successfully constructed (p = 0.000).

## Secondary outcomes

For secondary outcomes, the guidewire insertion time ($85.3 \pm 26.1$ vs $97.2 \pm 47.2$), total cannulation time ($215.8 \pm 40.8$ vs $244.5 \pm 61.5$) and the incidence of mechanical complications

**Table 2. Characteristics of subjects.**

| Parameters | Intervention groups | |
|---|---|---|
| | MCSL (n = 95) | OA-IP (n = 95) |
| Age (year) | $56.4 \pm 11.2$ | $55.1 \pm 10.7$ |
| Male sex n (%) | 35 (36.8) | 43 (45.3) |
| BMI (kg/m$^2$) | $23.6 \pm 3.6$ | $23.2 \pm 2.9$ |
| Length of the neck (cm) | $11.1 \pm 1.2$ | $11.4 \pm 1.4$ |
| Area of IJV (mm$^2$) | $103.4 \pm 53.6$ | $105.8 \pm 58.7$ |
| Depth of IJV (mm) | $8.6 \pm 2.3$ | $8.3 \pm 1.9$ |
| Overlapping (%) | $61.3 \pm 35.2$ | $63.7 \pm 33.4$ |
| Safe puncture distance (mm) | $9.6 \pm 4.9$ | $10.2 \pm 5.7$ |
| ASA physical status n (%) | | |
| Grade I/II | 75 (78.9) | 76 (80) |
| Grade III | 20 (21.1) | 19 (20) |
| Fasting time (h) | $16.6 \pm 3.0$ | $16 \pm 3.7$ |
| Bowel preparation n (%) | 49 (51.6) | 45 (47.4) |
| Preoperative infusion volume (ml) | 250 (0–500) | 250 (0–600) |
| Type of central venous tube n (%) | | |
| Single lumen tube (5 French) | 86 (90.5) | 88 (92.6) |
| Double lumen tube (7 French) | 9 (9.5) | 7 (7.4) |

Data shown as mean ± SD, median (IQR), or number and %. IJV, internal jugular vein; MCSL, modified combined short and long axis method; OA-IP, oblique axis in-plane method.

**Table 3. Comparison of procedure variables in full analysis population and per-protocol population.**

| Data Set | Parameters | Intervention groups | | Difference | |
|---|---|---|---|---|---|
| | | MCSL | OA-IP | Mean | 95% CI |
| FAS | **Included patients** | **95** | **95** | | |
| | First needle pass without PVWP n (%) | 85(89.5) | 81(85.3) | 4.2 | -5.2, 13.6 |
| | First successful needle pass n (%) | 85(89.5) | 86(90.5) | -1.1 | -9.6, 7.5 |
| | Second successful needle pass n (%) | 92 (96.8) | 94 (98.9) | -2.1 | -6.2, 2.0 |
| | Total successful needle pass n (%) | 92(96.8) | 95(100) | -3.2 | -6.7, 0.4 |
| | Visualization of needle n (%) | 91(95.8) | 94(98.9) | -3.2 | -7.7, 1.4 |
| | Probe placement difficulty n (%) | 0 (0) | 0 (0) | 0 | 0, 0 |
| | Total mechanical complications n (%) | 3 (3.2) | 10 (10.5) | -7.4 | -14.5, -0.3 |
| | PVWP n (%) | 1 (1.1) | 7 (7.4) | -6.3 | -12.0, -0.7 |
| | EJV puncture n (%) | 0 (0) | 3 (3.2) | -3.2 | -6.7, -0.4 |
| | Arterial puncture n (%) | 1 (1.1) | 0 (0) | 1.1 | -1.0, 3.1 |
| | Hematoma n (%) | 1 (1.1) | 0 (0) | 1.1 | -1.0, 3.1 |
| | Nerve damage n (%) | 0 (0) | 0 (0) | 0 | 0, 0 |
| | Thyroid injury n (%) | 0 (0) | 0 (0) | 0 | 0, 0 |
| | Pneumothorax n (%) | 0 (0) | 0 (0) | 0 | 0, 0 |
| | Hemothorax n (%) | 0 (0) | 0 (0) | 0 | 0, 0 |
| PPS | **Included patients** | **92** | **95** | | |
| | Guidewire insertion time (s) | 85.3 ± 26.1 | 97.2 ± 47.2 | -12.0 | -23.0, -0.9 |
| | Total cannulation time (s) | 215.8 ± 40.8 | 244.5 ± 61.5 | -24.8 | -41.1, -8.5 |
| | Needle attempts (n) | 1 (1–1) | 1 (1–1) | 0 | -0.1, 0.1 |

Data shown as mean ± SD, median (IQR), or number and %. MCSL, modified combined short and long axis method; OA-IP, oblique axis in-plane method; FAS, Full Analysis Set; PVWP, posterior vessel wall puncture; EJV, external jugular vein; PPS, Per Protocol Set.

(10.5% vs 3.2%) were less in the MCSL group than in the OA-IP group (P = 0.033, 0.003 and 0.044, respectively). Overall catheterization success rate of the two methods was 98.4%, and the difference between the groups was not statistically significant. No significant differences were discovered between the groups in needle attempts, incidence of probe placement difficulty, and visualization of needle (Table 3). Three IJVC cases (MCSL group) were identified unsuccessful due to needle attempts more than 3 times (n = 2) or arterial puncture (n = 1).

PVWP was the most frequently detected complication, with a higher tendency in the OA-IP group compare with MCSL group (7.4% vs 1.1%), but the difference was not statistically significant (P = 0.071). EJV puncture occurred only in the OA-IP group (n = 3). Except for one case of arterial puncture in MCSL group, no pneumothorax or other serious complications were found in either group.

## Discussion

In this randomized controlled clinical trial comparing two ultrasound-guided Technologies (MCSL and OA-IP) for right IJVC, we found that MCSL was noninferior to OA-IP in the first needle pass without PVWP, and was associated with less procedure time and fewer mechanical complications compared with OA-IP. Our data suggest that these two ultrasound-guided methods are equally effective in adult patients undergoing right IJVC.

OA-IP has been increasingly recommended in recent years for performing ultrasound-guided IJVC [15, 25, 31, 32]. Our findings regarding the number of needle pass, overall puncture success rate, first puncture success rate, and total cannulation time in OA-IP were

# Forest map about first needle pass without PVWP

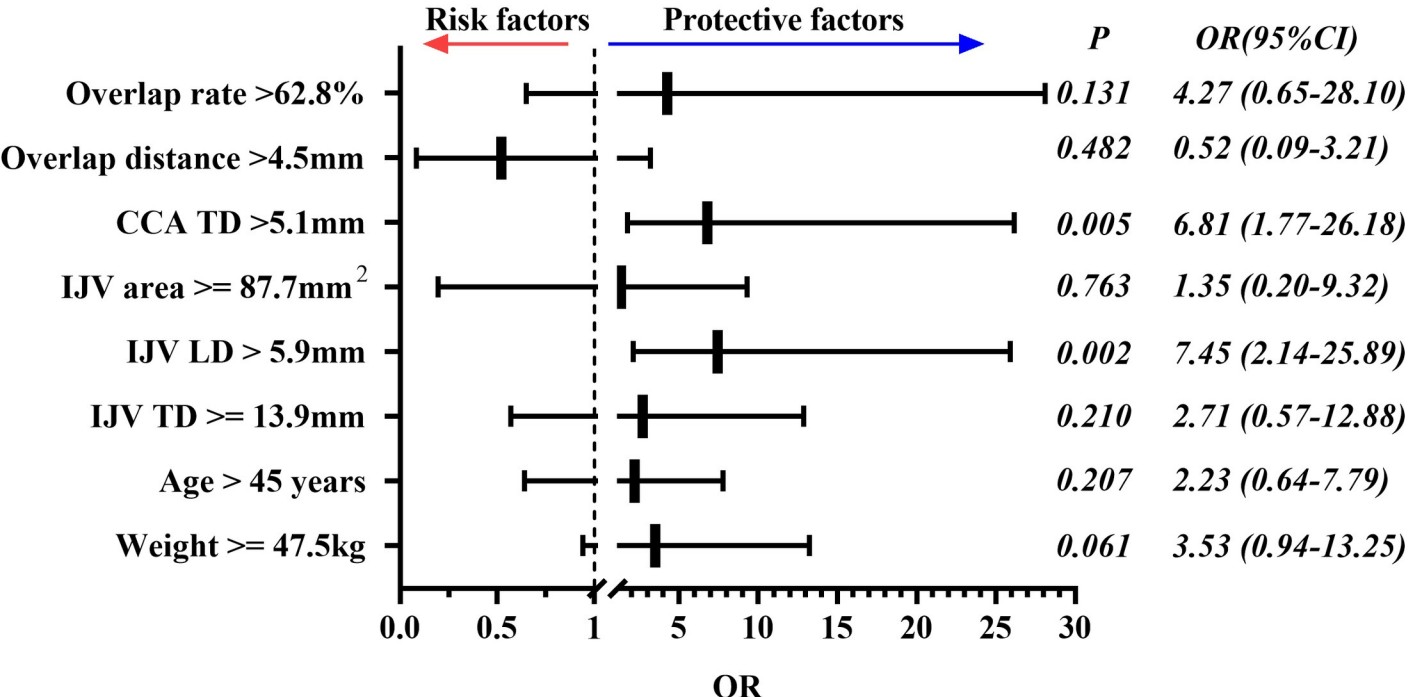

**Fig 3. Forest map about first needle pass without PVWP.** Cut-off values of each continuous quantitative data were determined by ROC curve test (S2 Table). PVWP, posterior vessel wall puncture; CCA, common carotid artery; TD, transverse diameter; IJV, internal jugular vein; LD, Longitudinal diameter.

consistent with previous studies [25, 31, 32]. Notably, our study introduced a stricter metric, the first needle pass without PVWP, not reported previously. Despite this, our success rate surpassed that of previous studies [15], likely due to our operator's extensive point-of-care US experience, having performed over 500 US-guided IJVC procedures. This suggests that the operator's experience may also affect the puncture success rate, warranting further clinical research to validate this conclusion.

In our study, OA-IP approach exhibited a higher incidence of PVWP compared to prior studies [25, 31, 32]. This divergence might be attributed to previous studies not assessing or considering PVWP as a complication, despite evidence suggesting its high incidence and potential impact on patient outcomes [12, 30, 33, 34]. We observed that the OA-IP method maintained a 45˚ angle between the puncture needle and the blood vessel during the procedure, with a narrower needle insertion space than the MCSL method. This difference may explain the higher PVWP incidence associated with the OA-IP method compared to the MCSL approach. We also observed three EJV puncture in the OA-IP group, which may be attributed to the proximity of the puncture site to the EJV in the lateral oblique axis method.

MCSL combines the strengths of the short-axis combined long-axis method and the modified short axis out of plane approach, and is currently a new method that has not been fully studied [16, 20]. Like OA-IP, MCSL harnesses the advantages of the long-axis in-plane technique and the short-axis view, resulting in a comparable puncture success rate to OA-IP.

The shorter guidewire insertion time and total cannulation time observed in the MCSL group in our study may be attributed to the operator's discomfort operation posture while

performing IJVC using the OA-IP technique. Furthermore, the OA-IP method necessitates additional time during the CVC procedure to prevent inadvertent puncture of the nearby EJV. In our research, 3 failed cases were found in the MCSL group. We considered that the main reason for puncture failure was that the ultrasound shadows was blurred and the puncture point cannot be accurately located. Future research aimed at improving the imaging quality of the ultrasound shadows may further enhance the puncture success rate.

Through logistic regression analysis, we found that IJV longitudinal diameter and CCA transverse diameter were closely related to the first needle pass without PVWP. It's common sense that more collapsed the blood vessel, the smaller of the puncture space, which certainly lead to the more difficult on puncture and the greater possibility of the posterior wall puncture. In addition, the IJV longitudinal diameter is the only one of these three factors that we can intervene on. What factors affect the IJV longitudinal diameter and how to increase it requires further analysis and research. The mechanism behind the increased success rate of puncture with an increased carotid artery diameter is currently unknown and requires further analysis.

Our study has some limitations. There was only one operator in this study, and differences in his proficiency with the two procedures may influence the results of the trial. We only included adult patients for right internal jugular vein puncture, and it must be cautious to extrapolate to children and left internal jugular vein. Future studies, including novice operators, involving pediatric populations and the left internal jugular vein, are essential to certify our findings.

## Conclusions

The main purpose of our current study was to estimate the clinical effect of the MCSL approach for ultrasound-guided IJVC. We found that MCSL was comparable to OA-IP in terms of catheter placement success and was associated with less mechanical complications and shorter operating time. We consider that MCSL is a safe and effective approach to perform IJVC and that its use should be encouraged.

## Supporting information

**S1 Fig. ROC curve analysis for first needle pass without PVWP.** ROC, receiver operating characteristic; PVWP, posterior vessel wall puncture; AUC, Area Under the Curve; CCA, Common carotid artery; IJV, internal jugular vein; BMI, Body Mass Index.
(TIF)

**S1 Text. Study protocol (original language).**
(PDF)

**S2 Text. Study protocol (translation).**
(PDF)

**S1 Table. CONSORT statement 2010 - Checklist for Non-inferiority and equivalence trials.**
(DOCX)

**S2 Table. Calculating optimal cutoff values using ROC curve for continuous variables.**
(XLSX)

## Acknowledgments

We thank the technical support of Qianyun Pang from Chongqing University Cancer Hospital.

## Author Contributions

**Conceptualization:** Jia-Xi Tang, Ling Wang, Hongliang Liu, Fang Xu.

**Data curation:** Jia-Xi Tang.

**Formal analysis:** Ling Wang.

**Funding acquisition:** Jia-Xi Tang, Hongliang Liu.

**Investigation:** Jia-Xi Tang, Xixi Tang, Mengxiao Liu.

**Methodology:** Jia-Xi Tang, Ju Ouyang, Xixi Tang, Fang Xu.

**Project administration:** Jia-Xi Tang.

**Resources:** Jia-Xi Tang.

**Supervision:** Hongliang Liu, Fang Xu.

**Validation:** Jia-Xi Tang, Ling Wang, Ju Ouyang, Xixi Tang, Mengxiao Liu, Hongliang Liu, Fang Xu.

**Writing – original draft:** Jia-Xi Tang.

**Writing – review & editing:** Hongliang Liu, Fang Xu.

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
