## [Decision Letter · Decision Letter 0]

21 Sep 2023

PONE-D-23-27163Modified combined short and long axis method versus oblique axis method in adult patients undergoing right internal jugular vein cannulation: A randomised controlled non-inferiority studyPLOS ONE

Dear Dr. Tang,

Thank you for submitting your manuscript to PLOS ONE. After careful consideration, we feel that it has merit but does not fully meet PLOS ONE’s publication criteria as it currently stands. Therefore, we invite you to submit a revised version of the manuscript that addresses the points raised during the review process.

Please revise according to the comments provided by the 3 reviewers.

We look forward to receiving your revised manuscript.

Kind regards,

Luigi La Via

Academic Editor

PLOS ONE

Journal Requirements:

Reviewers' comments:

Reviewer's Responses to Questions

**Comments to the Author**

1. Is the manuscript technically sound, and do the data support the conclusions?

Reviewer #1: Yes

Reviewer #2: Yes

Reviewer #3: Yes

2. Has the statistical analysis been performed appropriately and rigorously? 

Reviewer #1: I Don't Know

Reviewer #2: Yes

Reviewer #3: Yes

3. Have the authors made all data underlying the findings in their manuscript fully available?

Reviewer #1: Yes

Reviewer #2: Yes

Reviewer #3: Yes

4. Is the manuscript presented in an intelligible fashion and written in standard English?

Reviewer #1: No

Reviewer #2: No

Reviewer #3: Yes

5. Review Comments to the Author

Reviewer #1: Thank you for allowing me to review "Modified combined short and long axis method versus oblique axis method in adult patients undergoing right internal jugular vein cannulation: A randomised controlled non-inferiority study" by Jia-Xi Tang et al.

The authors compared first needle pass without posterior vessel wall puncture between modified combined short and long axis method (MCSL) and oblique axis in-plane method (OA-IP). The first needle pass without posterior vessel wall puncture was not significantly different between the two groups.

This is an interesting study. However, I may be mistaken, but there may be a critical problem.

My suggestions are as follows.

American and British English are mixed.

Introduction

“the operator's puncture posture is very awkward”

I have never used the OA-IP method. Please explain how it is awkward.

Methods

Intervention

Only one operator performed all IJVC. This is a very serious limitation.

Study endpoints and sample size calculation ”We estimated that 85 patients per group would supply 90% power at a level of 0.025 to compare the MCSL approach with the OA-IP method.“

Do you mean “at an α level” ?

Furthermore, using the numbers you indicated and doing a sample size calculation, I came up with 1615 per group, even as a one-tailed test. I am no expert in statistics, but I think the journal should check to see if the author’s sample size calculation is correct.

Primary outcome

“The difference in rate (MCSL vs OA-IP) was 4.2% (95% CI: -5.5-13.6), which confirmed the 207 non-inferiority with the margin of -8%.”

Since the 95% confidence interval includes a margin, I don't think non-inferiority has been proven.

Reference11

Watanabe K, Tokumine J, Lefor AK, et al.

Reviewer #2: Thank you for the opportunity to revise this interesting randomized trial by Tang et al. on the comparison between combined short and long axis method and oblique axis method in adult patients undergoing right internal jugular vein cannulation. The paper is sound and original. However, there are some issues that need to be addressed:

- I believe there is an issue with the reference style, as I don't think they should be in superscript in the manuscript.

- Line 48-49. Authors should also add that ultrasound guided cannulation has proven to be safer and better than landmark technique regardless the cannulation site (doi: 10.1097/CCM.0000000000005819 - doi: 10.1002/14651858.CD01144). Please briefly discuss and cite these 2 references.

- Line 56. Why is the operator's puncture posture very awkward? Please specify.

- The manuscript must be edited for English, as there are some minor mistakes, like "have" instead of "has" in line 73.

- Line 140. Why you did not use triple or quadruple lumen cathethers? Please specify the caliber in French of the catheters used.

- Please increase the text size in all the Tables presented to increase readability.

Reviewer #3: I am pleased to commend you on your research paper titled "Modified Combined Short and Long Axis Method Versus Oblique Axis Method in Adult Patients Undergoing Right Internal Jugular Vein Cannulation: A Randomized Controlled Non-Inferiority Study." Your study exhibits a robust scientific approach and has made a valuable contribution to the field of medical procedures. The meticulous design of your randomized controlled non-inferiority study reflects a high level of scientific rigor. Your choice of research methodology and the focus on a topic as important as vein cannulation in adult patients demonstrate your dedication to improving medical practices. In conclusion, I believe that your study has the potential to make a positive impact in the field of medical procedures and should be accepted for publication with the suggested revisions. While your research is indeed noteworthy, I would recommend minor revisions and an English language edit to enhance the clarity and readability of the manuscript. These revisions will ensure that your valuable findings can be effectively communicated to a broader audience.

Introduction: The introduction is the weakest part of the manuscript pleas consider English language edit

47 there is a growing body of evidence supporting the use of ultrasound-guided subclavian access: RCTs and meta-analysis. It might be a preferred option for ICU patients. Consider providing information for the reader about new evidence and different characteristics for CVC in OR and ICU population.

57 please restrain from using colloquial words like awkward and use scientific descriptions.

58 it would be more appropriate to use structures rather than organs

Methods:

Sound scientific methodology with study register and CONSORT statement.

patients were randomized appropriately.

Please move sample size calculation to methods section

Intervention:

Detailed description of the intervention.

Study endpoint:

Please define probe placement difficulty, needle visualization

Secondary outcomes:

Would it be possible to differentiate mechanical complications? please provide n for artery puncture, hematoma, pneumothorax etc. It will help to include your research in future meta-analysis

Discussion:

There was a mean difference of 25 s of cannulation time. Is it clinically relevant difference?

The authors use a rigorous definition of first needle pass without PVWP. The success rate is exceptionally high do you think that your results would be replicable with less experienced operators? Do you consider MCSL an advanced US technique? What would be the learning curve ? Interested to hear your opinion

Discussion section could shortened and more focused

6. PLOS authors have the option to publish the peer review history of their article (what does this mean?). If published, this will include your full peer review and any attached files.

Reviewer #1: No

Reviewer #2: No

Reviewer #3: No

---

## [Author Response · Author response to Decision Letter 0]

29 Sep 2023

Jiaxi Tang MD

First Affiliated Hospital of Chongqing Medical University, Chongqing, China

Chongqing University Cancer Hospital, Chongqing 400030, China

E-mail: tangjiaxi1029@126.com

Sep 25, 2023

RE: Manuscript ID: PONE-D-23-27163

Dear Editor and Reviewers，

We would like to thank the editor for giving us a chance to resubmit the paper, and also thank the reviewers for giving us constructive suggestions which would help us both in English and in depth to improve the quality of the paper. Here we submit a revised version of our manuscript with the title “Modified combined short and long axis method versus oblique axis method in adult patients undergoing right internal jugular vein cannulation: A randomized controlled non-inferiority study”, which has been modified according to the reviewers’ suggestions. Efforts were also made to correct the mistakes and improve the English of the manuscript. We track all the changes in the revised manuscript.

Sincerely yours,

Jiaxi Tang MD

The following is a point-to-point response to editor and the three reviewers’ comments.

Editor:

Specific Comments:

Editor: When submitting your revision, we need you to address these additional requirements.

Answer: Thank you for your guidance and assistance in our submission process. Your efficient and responsible work is admirable. We have checked the format of the manuscript again to confirm that our manuscript meets PLOS ONE's style requirements.

Reviewer #1:

Specific Comments:

Reviewer #1: Thank you for allowing me to review "Modified combined short and long axis method versus oblique axis method in adult patients undergoing right internal jugular vein cannulation: A randomised controlled non-inferiority study" by Jia-Xi Tang et al.

The authors compared first needle pass without posterior vessel wall puncture between modified combined short and long axis method (MCSL) and oblique axis in-plane method (OA-IP). The first needle pass without posterior vessel wall puncture was not significantly different between the two groups.

This is an interesting study. However, I may be mistaken, but there may be a critical problem.

My suggestions are as follows.

American and British English are mixed.

Introduction

“the operator's puncture posture is very awkward”

I have never used the OA-IP method. Please explain how it is awkward.

Methods

Intervention

Only one operator performed all IJVC. This is a very serious limitation.

Study endpoints and sample size calculation ”We estimated that 85 patients per group would supply 90% power at a level of 0.025 to compare the MCSL approach with the OA-IP method.“

Do you mean “at an α level” ?

Furthermore, using the numbers you indicated and doing a sample size calculation, I came up with 1615 per group, even as a one-tailed test. I am no expert in statistics, but I think the journal should check to see if the author’s sample size calculation is correct.

Primary outcome

“The difference in rate (MCSL vs OA-IP) was 4.2% (95% CI: -5.5-13.6), which confirmed the non-inferiority with the margin of -8%.”

Since the 95% confidence interval includes a margin, I don't think non-inferiority has been proven.

Reference11

Watanabe K, Tokumine J, Lefor AK, et al.

Answer: Thank you for the comments on the paper. Thanks for your affirmation of this article. Thank you for your pertinent comments, which have greatly helped to improve the quality and rigor of the article. We have made corresponding revisions according to your comments.

1. American and British English are mixed.

Answer: Thank you for the comments on the paper, which will greatly help to improve the accuracy of the English use of the manuscript. We have carefully checked the error of the misuse of American and British English and also corrected errors in syntax and text formatting. (Line 3, 27, 29, 34, 36, 57, 59, 63, 76, 84, 87-89, 104, 107, 122, 134, 135, 163, 177, 178, 180, 186, 187, 190, 196, 197, 215, 219, 229, 243, 248, 250, 251, 258, 300, 366, 371, 521, 522)

2. Introduction

“the operator's puncture posture is very awkward”

I have never used the OA-IP method. Please explain how it is awkward.

Answer: Thank you for the comments and reminders on the paper, which will increase the reader's understanding of the document. We have added a description of the OA-IP operation details and why this operation is awkward. (Line 64-68)

 In addition, we have uploaded two pictures of OA-IP puncture operation to help reviewers understand the OA-IP puncture operation method. (see attachment1)

3. Methods

Intervention

Only one operator performed all IJVC. This is a very serious limitation.

Answer: Thank you for your insights and pointing out the shortcomings of our research, which is very enlightening for us. In order to reduce the difference in technical level of operators, we adopted a single-person operation design. This design does have limitations, and we also stated this limitation in the manuscript. We will use more reasonable designs in future studies to reduce the impact of such biases on experimental results. Thanks again to the reviewers for their sincere comments. (Line 116, 117, 351, 352)

4. Study endpoints and sample size calculation ”We estimated that 85 patients per group would supply 90% power at a level of 0.025 to compare the MCSL approach with the OA-IP method.“Do you mean “at an α level” ? 

Furthermore, using the numbers you indicated and doing a sample size calculation, I came up with 1615 per group, even as a one-tailed test. I am no expert in statistics, but I think the journal should check to see if the author’s sample size calculation is correct.

Answer: Thank you for your comments and questions, which are necessary for scientific rigor. We used PASS software (version 15) to calculate the sample size of the study, using the sample size calculation formula for non-inferiority studies. Please see attachment 2 for the specific calculation process. In addition, we thank you for correcting the problems in our presentation, indeed at the α level. We have corrected the issue. By calculating again, we found that there is a slight difference from the previous calculation. We have modified it based on the latest calculation, and there is no difference in the total sample size. (Line 188, 189)

5. Primary outcome

“The difference in rate (MCSL vs OA-IP) was 4.2% (95% CI: -5.5-13.6), which confirmed the non-inferiority with the margin of -8%.”

Since the 95% confidence interval includes a margin, I don't think non-inferiority has been proven.

Reference11

Watanabe K, Tokumine J, Lefor AK, et al.

Answer: Thank you for your comments. We are sorry that our writing may have misled the reviewers. In our study, the mean difference in Primary outcome was 4.2% in MCSL vs OA-IP, and the 95% confidence interval was -5.5% to 13.6%. This confidence interval did not include -8%, so we believe that non-inferiority can be proved. We have standardized the use of hyphens so that hyphens can be distinguished from minus signs (Line 29, 36, 123, 200, 217, 219, 224, 233, 251).

Reviewer #2:

Specific Comments:

Reviewer #2: Thank you for the opportunity to revise this interesting randomized trial by Tang et al. on the comparison between combined short and long axis method and oblique axis method in adult patients undergoing right internal jugular vein cannulation. The paper is sound and original. However, there are some issues that need to be addressed:

- I believe there is an issue with the reference style, as I don't think they should be in superscript in the manuscript.

- Line 48-49. Authors should also add that ultrasound guided cannulation has proven to be safer and better than landmark technique regardless the cannulation site (doi: 10.1097/CCM.0000000000005819 - doi: 10.1002/14651858.CD01144). Please briefly discuss and cite these 2 references.

- Line 56. Why is the operator's puncture posture very awkward? Please specify.

- The manuscript must be edited for English, as there are some minor mistakes, like "have" instead of "has" in line 73.

- Line 140. Why you did not use triple or quadruple lumen cathethers? Please specify the caliber in French of the catheters used.

- Please increase the text size in all the Tables presented to increase readability.

Answer: Thank you for your comments and valuable suggestions, which will greatly help to improve the quality and readability of the manuscript. We have revised it according to the reviewer's comments. Thank you again for your review and hard work.

1. I believe there is an issue with the reference style, as I don't think they should be in superscript in the manuscript.

Answer: Yes, your opinions inspired us and we revised the manuscript accordingly. (Line 49, 51, 54, 57, 58, 60, 63, 74, 87, 89, 135, 266, 268, 272, 280, 285, 302)

2. Line 48-49. Authors should also add that ultrasound guided cannulation has proven to be safer and better than landmark technique regardless the cannulation site (doi: 10.1097/CCM.0000000000005819 - doi: 10.1002/14651858.CD01144). Please briefly discuss and cite these 2 references.

Answer: Thank you for your references. These two articles are very valuable, which are very important to enrich and clarify the point of view of our manuscript. We have briefly discuss and cite these 2 references. (Line 57, 416-424).

3. Line 56. Why is the operator's puncture posture very awkward? Please specify.

Answer: Thank you for the comments and reminders on the paper, which will increase the reader's understanding of the document. We have added a description of the OA-IP operation details and why this operation is awkward (Line 64-68).

 In addition, we have uploaded two pictures of OA-IP puncture operation to help reviewers understand the OA-IP puncture operation method. (see attachment1)

4. The manuscript must be edited for English, as there are some minor mistakes, like "have" instead of "has" in line 73.

Answer: Thank you for your careful and responsible review and help, which has greatly helped to improve the quality and readability of the manuscript. We have carefully checked for similar errors. Thanks again. (Line 57, 59, 87, 107, 300, 366)

5. Line 140. Why you did not use triple or quadruple lumen cathethers? Please specify the caliber in French of the catheters used.

Answer: We are sorry that we did not use triple or quadruple lumen cathethers, since our institution did not purchase these types of catheter. 

 Thank you for your reminder and guidance. Specifying the size of the catheter makes the writing of the manuscript more standardized, easier for readers to understand, and increases the comparability of different studies. (Line 156, 219)

6. Please increase the text size in all the Tables presented to increase readability.

Answer: Thank you for your helpful advice, which are very important to improve the readability of the article. We have increased the font size in all the tables to 12. (Line 104, 219, 233)

Reviewer #3:

Specific Comments:

Reviewer #3: I am pleased to commend you on your research paper titled "Modified Combined Short and Long Axis Method Versus Oblique Axis Method in Adult Patients Undergoing Right Internal Jugular Vein Cannulation: A Randomized Controlled Non-Inferiority Study." Your study exhibits a robust scientific approach and has made a valuable contribution to the field of medical procedures. The meticulous design of your randomized controlled non-inferiority study reflects a high level of scientific rigor. Your choice of research methodology and the focus on a topic as important as vein cannulation in adult patients demonstrate your dedication to improving medical practices. In conclusion, I believe that your study has the potential to make a positive impact in the field of medical procedures and should be accepted for publication with the suggested revisions. While your research is indeed noteworthy, I would recommend minor revisions and an English language edit to enhance the clarity and readability of the manuscript. These revisions will ensure that your valuable findings can be effectively communicated to a broader audience.

Introduction: The introduction is the weakest part of the manuscript pleas consider English language edit

47 there is a growing body of evidence supporting the use of ultrasound-guided subclavian access: RCTs and meta-analysis. It might be a preferred option for ICU patients. Consider providing information for the reader about new evidence and different characteristics for CVC in OR and ICU population.

57 please restrain from using colloquial words like awkward and use scientific descriptions.

58 it would be more appropriate to use structures rather than organs

Methods:

Sound scientific methodology with study register and CONSORT statement.

patients were randomized appropriately.

Please move sample size calculation to methods section

Intervention:

Detailed description of the intervention.

Study endpoint:

Please define probe placement difficulty, needle visualization

Secondary outcomes:

Would it be possible to differentiate mechanical complications? please provide n for artery puncture, hematoma, pneumothorax etc. It will help to include your research in future meta-analysis

Discussion:

There was a mean difference of 25 s of cannulation time. Is it clinically relevant difference?

The authors use a rigorous definition of first needle pass without PVWP. The success rate is exceptionally high do you think that your results would be replicable with less experienced operators? Do you consider MCSL an advanced US technique? What would be the learning curve ? Interested to hear your opinion

Discussion section could shortened and more focused

Answer: Thank you for your review, which makes a huge contribution to the dissemination of science. Thank you very much for your recognition and valuable comments on our paper. We will improve our manuscript based on your comments.

1. Introduction: The introduction is the weakest part of the manuscript pleas consider English language edit

Answer: Thank the reviewer for the comments. We have checked the manuscript for language errors and inauthentic expressions, and we have also asked native English speakers to help check the manuscript for language problems, hoping to increase the readability and accuracy of the manuscript. ( Line 48-57, 59, 87).

2. 47 there is a growing body of evidence supporting the use of ultrasound-guided subclavian access: RCTs and meta-analysis. It might be a preferred option for ICU patients. Consider providing information for the reader about new evidence and different characteristics for CVC in OR and ICU population.

Answer: Thank you for your constructive comments, which are very important for readers to have a comprehensive understanding of the background knowledge of this research field. We have added new advances related to ultrasound-guided subclavian venipunctures, and different characteristics for CVC in OR and ICU population. (Line 48-53).

3. 57 please restrain from using colloquial words like awkward and use scientific descriptions.

Answer: Thank the reviewer for the comments. We have replaced colloquial words with more scientific expressions (Line 66-69, 306, 312).

4. 58 it would be more appropriate to use structures rather than organs

Answer: Thank you for your guidance and help, which is very important to improve the quality of the manuscript. We have made changes according to your guidance (Line 70).

5. Methods:

Sound scientific methodology with study register and CONSORT statement.

patients were randomized appropriately.

Please move sample size calculation to methods section

Answer: Thank you for your affirmation and comments. This may be because we did not use the number subheading, leading to misunderstanding. According to the requirements of the magazine, we have used font size to distinguish between titles and sub-titles. In our manuscript, the Sample size calculation font is 16, which belongs to the Methods section (Line 82, 157).

6. Intervention:

Detailed description of the intervention.

Answer: Thank you for your suggestions and comments. In the intervetion section, we describe the information of the equipment used in this study, the information of the operator, the content of data collection, and the operation steps of the two puncture methods. If there are any gaps in our description, please kindly advise the reviewer. Thank you (Line 115-156).

7. Study endpoint:

Please define probe placement difficulty, needle visualization

Answer: Thank you for your comments and valuable suggestions, which are of great help to readers in understanding our research. We have refined the content of this section (Line 161-163, 168, 169). 

8. Secondary outcomes:

Would it be possible to differentiate mechanical complications? please provide n for artery puncture, hematoma, pneumothorax etc. It will help to include your research in future meta-analysis

Answer: Thank the reviewer for the comments and your suggestion is greatly appreciated. Maybe we haven't stated it clearly, but we have distinguished the different complications and organized these results in Table 3. To prevent ambiguity, we have made some minor changes to the table 3. The total mechanical complications in the table 3 is the sum of all complications (Line 233, see Table 3). 

9. Discussion:

There was a mean difference of 25 s of cannulation time. Is it clinically relevant difference?

Answer: Thank you for your review and for asking this meaningful question. The average total cannulation time is about 200 seconds. Compared with the OA-IP group, the MCSL group saves about 25 seconds in the operation of each patient, about 1/10 of the time. Overall, it will improve work efficiency and have a positive effect on busy clinical work. Because this indicator is not the main indicator we care about, the clinically meaningful difference that this indicator needs to achieve is not preset. 

10. The authors use a rigorous definition of first needle pass without PVWP. The success rate is exceptionally high do you think that your results would be replicable with less experienced operators?

Answer: Thank the reviewer for the comments. Our study chose a more stringent metric, and we achieved a fairly high success rate. For this reason, which we mentioned in the Discussion section, our operators were very skilled practitioners and had become proficient in both puncture methods before conducting this study. If it were performed by a novice who was completely unfamiliar with the two methods, or even unfamiliar with ultrasound-guided vascular intervention, perhaps the success rate would not be so high. Fortunately, both ultrasound-guided methods have a flat learning curve and can be mastered quickly. (Line 268-274). 

11. Do you consider MCSL an advanced US technique? What would be the learning curve ? Interested to hear your opinion. 

Answer: Thank you for your review and your interest in this technology and research, we are very willing to share it. First and foremost, we firmly believe that MCSL is an invaluable technique that amalgamates the strengths of both the short axis and the long axis approaches. Furthermore, it leverages ultrasound artifacts for precise puncture site localization, thereby ensuring puncture accuracy. Additionally, the learning curve associated with this method is relatively gentle, making it particularly beginner-friendly. 

 We were under the impression that mastering this technique would require only a few cases. Unfortunately, we did not collect data regarding the learning process for this technique. In future research, we anticipate well-designed studies to assess the learning curve for novices, with the aim of obtaining accurate learning curve data. 

12. Discussion section could shortened and more focused

Answer: Thank the reviewer for the comments and your suggestion is greatly appreciated. We have condensed the discussion section by removing redundant portions (Line 264-341).

---

## [Decision Letter · Decision Letter 1]

21 Nov 2023

PONE-D-23-27163R1Modified combined short and long axis method versus oblique axis method in adult patients undergoing right internal jugular vein cannulation: A randomised controlled non-inferiority studyPLOS ONE

Dear Dr. Tang,

Thank you for submitting your manuscript to PLOS ONE. After careful consideration, we feel that it has merit but does not fully meet PLOS ONE’s publication criteria as it currently stands. Therefore, we invite you to submit a revised version of the manuscript that addresses the points raised during the review process.

Please revise according to the comments provided by reviewer 1 and reviewer 4

We look forward to receiving your revised manuscript.

Kind regards,

Luigi La Via

Academic Editor

PLOS ONE

Additional Editor Comments:

Please revise according to the comments provided by reviewer 1 and reviewer 4

Reviewers' comments:

Reviewer's Responses to Questions

**Comments to the Author**

1. If the authors have adequately addressed your comments raised in a previous round of review and you feel that this manuscript is now acceptable for publication, you may indicate that here to bypass the “Comments to the Author” section, enter your conflict of interest statement in the “Confidential to Editor” section, and submit your "Accept" recommendation.

Reviewer #1: All comments have been addressed

Reviewer #2: (No Response)

Reviewer #3: All comments have been addressed

Reviewer #4: (No Response)

2. Is the manuscript technically sound, and do the data support the conclusions?

Reviewer #1: Yes

Reviewer #2: (No Response)

Reviewer #3: Yes

Reviewer #4: Partly

3. Has the statistical analysis been performed appropriately and rigorously? 

Reviewer #1: Yes

Reviewer #2: (No Response)

Reviewer #3: Yes

Reviewer #4: No

4. Have the authors made all data underlying the findings in their manuscript fully available?

Reviewer #1: Yes

Reviewer #2: (No Response)

Reviewer #3: Yes

Reviewer #4: No

5. Is the manuscript presented in an intelligible fashion and written in standard English?

Reviewer #1: Yes

Reviewer #2: (No Response)

Reviewer #3: Yes

Reviewer #4: No

6. Review Comments to the Author

Reviewer #1: Thank you for allowing me to review "Modified combined short and long axis method versus oblique axis method in adult patients undergoing right internal jugular vein cannulation: A randomised controlled non-inferiority study" by Jia-Xi Tang et al.

The authors addressed the reviewer’s comments. I confirmed that the sample size calculation was correct and that non-inferiority was proved.

I have only one comment.

Limitation

The authors stated that “The operator in this study was an attending physician, and his proficiency in both procedures may influence the results of the trial.” in the limitation section.

However, I think it should be emphasized that there is only one operator.

Reviewer #2: Dear authors, all the comments were successfully addressed. I believe the paper is now ready for acceptance.

Reviewer #3: Thank you for your prompt revisions and thorough attention to the reviewers' comments. I have carefully reviewed the updated manuscript, and I am pleased to confirm that all the suggested changes have been successfully addressed.

I have no further comments at this time, and I appreciate your diligence in improving the manuscript. Your efforts have significantly strengthened the overall quality of the work.

I look forward to seeing the final version of your manuscript.

Reviewer #4: The authors present the results of a randomised controlled non-inferiority study with a two arm parallel group design to compare MCSL method with OAIP method with repect to first needle pass without posterior vessel wall puncture (PVWP) in patients with IJVC patients. They concluded that MCSL is non-inferior to OA-IP.

In general, the presentation needs revision with respect to clarification of sample size justification, and non-inferiority analysis.

• Please provide a CONSORT checklist for non-inferiority studies (PLoS Med. 2010;7(3):e1000251. PMID: 2035206). (Please address every aspect of the checklist within the paper.)

• (formulate of non-inferiority hypothesis): Please formulate the specific non-inferiority hypothesis. The description does not specify whether it should be proven that MCSL is non-inferior to OA-IP or vice versa in relation to the non-inferiority margin of 8%.

• (analysis of primary endpoint variable (PeV): To proof the non-inferiority hypothesis a one-sided 95% confidence interval should be given. This should be calculated for the ITT as well as PP population to confirm rejection of the null hypothesis. (see abstract and result section)

L28: Please rephrase: ...primary outcome is the event of first .....

L34: see analysis of PeV

L41: As the register is not accessible, please provide a English translated confirmed entry of data.

L70: see formulate of non-inferiority hypothesis

L72: Compare is misleading term here.

L76: Evaluator blinded is not clear. Please explain, who assessed the PeV and how blinding is reached.

L80: Use CONSORT for non-inferiority

L94: PBR(4) results in a high number of predictable allocations, here at least 47 up to 95 predictable allocations, in particular with implementation by sealed opaque envelops. This causes allocation bias and need to be discussed in the limitation section.

L99. Please explain the meaning and implementation process to blinding of monitors and stat analysts with respect to detection, performance and attrition bias.

L144ff: Please describe the time and measurement method of the PeV - PVWP. Here in particular the validity of the measurement methods, i.e. objective measurement methods (measurement free of knowledge about allocated treatment), needs to be described.

L171: I could not follow the sample size calculation. My program results in 186/group using the stated numbers. Please provide details of the input data, statistical rule for calculation of the confidence interval as main analysis and software program used for computation.

L173: Please justify the non-inferiority margin, e.g. by expected difference of standard to placebo.

L177: Please describe the analysis populations.

L186: Rephrase according to analysis of PeV.

L193: Skip statistical tests on baseline data, not meaningful in RCT's.

L196: Please skip data across group = overall data. Not meaningful.

L204: Significance test is not meaningful here. Use the one sided 95% confidence interval for argumentation and show how the CI is similar in ITT and PP.

L273: Avoid reporting of results in the discussion section.

L281: Limitation section should focus on operational and methodological aspects of the study. Include discussion of sample size justification, if needed and allocation bias.

7. PLOS authors have the option to publish the peer review history of their article (what does this mean?). If published, this will include your full peer review and any attached files.

Reviewer #1: **Yes: **Jun Takeshita

Reviewer #2: No

Reviewer #3: No

Reviewer #4: No

---

## [Author Response · Author response to Decision Letter 1]

27 Nov 2023

Jiaxi Tang MD

First Affiliated Hospital of Chongqing Medical University, Chongqing, China

Chongqing University Cancer Hospital, Chongqing 400030, China

E-mail: tangjiaxi1029@126.com

Nov 22, 2023

RE: Manuscript ID: PONE-D-23-27163R1

Dear Editor and Reviewers，

We would like to thank the editor for giving us a chance to resubmit the paper, and also thank the reviewers for giving us constructive suggestions which would help us both in depth and language to improve the quality of the paper. Here we submit a revised version of our manuscript with the title “Modified combined short and long axis method versus oblique axis method in adult patients undergoing right internal jugular vein cannulation: A randomized controlled non-inferiority study”, which has been modified according to the reviewers’ suggestions. We track all the changes in the revised manuscript.

Sincerely yours,

Jiaxi Tang MD

The following is a point-to-point response to editor and the four reviewers’ comments.

Editor:

Specific Comments:

Editor: Please include the following items when submitting your revised manuscript:

4. Please revise according to the comments provided by reviewer 1 and reviewer 4.

Answer: Thank you for your guidance and assistance in our submission process. Your efficient and responsible work is admirable. We will revise our manuscript according to the editor and reviewers and submit the revised manuscript.

Reviewer #1:

Specific Comments:

Reviewer #1: Thank you for allowing me to review "Modified combined short and long axis method versus oblique axis method in adult patients undergoing right internal jugular vein cannulation: A randomised controlled non-inferiority study" by Jia-Xi Tang et al.

The authors addressed the reviewer’s comments. I confirmed that the sample size calculation was correct and that non-inferiority was proved.

I have only one comment.

Limitation

The authors stated that “The operator in this study was an attending physician, and his proficiency in both procedures may influence the results of the trial.” in the limitation section.

However, I think it should be emphasized that there is only one operator.

Answer: Thank you for reviewing the manuscript again and for your efforts in improving the quality of our manuscript. We also appreciate your affirmation and your pertinent opinions. We have made corresponding revisions according to your comments.

1. Limitation

The authors stated that “The operator in this study was an attending physician, and his proficiency in both procedures may influence the results of the trial.” in the limitation section.

However, I think it should be emphasized that there is only one operator.

Answer: Thank you for the comments on the paper, which are very enlightening and accurately point out the limitation of our research design. We have made corresponding modifications in accordance with your comments to more clearly state the limitations of our research. (Line 299-301)

Reviewer #2:

Specific Comments:

Reviewer #2: Dear authors, all the comments were successfully addressed. I believe the paper is now ready for acceptance.

Answer: Thank you for your previous comments and valuable suggestions, which greatly help to improve the quality and readability of the manuscript. In addition, thank you for your recognition of our work and your praise for our research. Thank you again for your hard work.

Reviewer #3:

Specific Comments:

Reviewer #3: Thank you for your prompt revisions and thorough attention to the reviewers' comments. I have carefully reviewed the updated manuscript, and I am pleased to confirm that all the suggested changes have been successfully addressed.

I have no further comments at this time, and I appreciate your diligence in improving the manuscript. Your efforts have significantly strengthened the overall quality of the work.

I look forward to seeing the final version of your manuscript.

Answer: Thank you for your outstanding efforts and contributions to improving the quality of our manuscript, and for your recognition and appreciation of our work.

Reviewer #4:

Specific Comments:

Reviewer #4: The authors present the results of a randomised controlled non-inferiority study with a two arm parallel group design to compare MCSL method with OAIP method with repect to first needle pass without posterior vessel wall puncture (PVWP) in patients with IJVC patients. They concluded that MCSL is non-inferior to OA-IP.

In general, the presentation needs revision with respect to clarification of sample size justification, and non-inferiority analysis.

• Please provide a CONSORT checklist for non-inferiority studies (PLoS Med. 2010;7(3):e1000251. PMID: 2035206). (Please address every aspect of the checklist within the paper.)

• (formulate of non-inferiority hypothesis): Please formulate the specific non-inferiority hypothesis. The description does not specify whether it should be proven that MCSL is non-inferior to OA-IP or vice versa in relation to the non-inferiority margin of 8%.

• (analysis of primary endpoint variable (PeV): To proof the non-inferiority hypothesis a one-sided 95% confidence interval should be given. This should be calculated for the ITT as well as PP population to confirm rejection of the null hypothesis. (see abstract and result section)

L28: Please rephrase: ...primary outcome is the event of first .....

L34: see analysis of PeV

L41: As the register is not accessible, please provide a English translated confirmed entry of data.

L70: see formulate of non-inferiority hypothesis

L72: Compare is misleading term here.

L76: Evaluator blinded is not clear. Please explain, who assessed the PeV and how blinding is reached.

L80: Use CONSORT for non-inferiority

L94: PBR(4) results in a high number of predictable allocations, here at least 47 up to 95 predictable allocations, in particular with implementation by sealed opaque envelops. This causes allocation bias and need to be discussed in the limitation section.

L99. Please explain the meaning and implementation process to blinding of monitors and stat analysts with respect to detection, performance and attrition bias.

L144ff: Please describe the time and measurement method of the PeV - PVWP. Here in particular the validity of the measurement methods, i.e. objective measurement methods (measurement free of knowledge about allocated treatment), needs to be described.

L171: I could not follow the sample size calculation. My program results in 186/group using the stated numbers. Please provide details of the input data, statistical rule for calculation of the confidence interval as main analysis and software program used for computation.

L173: Please justify the non-inferiority margin, e.g. by expected difference of standard to placebo.

L177: Please describe the analysis populations.

L186: Rephrase according to analysis of PeV.

L193: Skip statistical tests on baseline data, not meaningful in RCT's.

L196: Please skip data across group = overall data. Not meaningful.

L204: Significance test is not meaningful here. Use the one sided 95% confidence interval for argumentation and show how the CI is similar in ITT and PP.

L273: Avoid reporting of results in the discussion section.

L281: Limitation section should focus on operational and methodological aspects of the study. Include discussion of sample size justification, if needed and allocation bias.

Answer: Thank you for your professional and valuable comments and substantial contribution to improving the quality and rigor of our manuscript. We have revised our manuscript based on your comments.

1. Please provide a CONSORT checklist for non-inferiority studies (PLoS Med. 2010;7(3):e1000251. PMID: 2035206). (Please address every aspect of the checklist within the paper.)

Answer: Thank the reviewer for the comments. We carefully read the CONSORT checklist for non-inferiority and presented the content required in the checklist into our manuscript, and we provided the CONSORT checklist for non-inferiority in the supporting document (S1 Table). We also cited the corresponding literature in the manuscript.( Line 82-84).

2. (formulate of non-inferiority hypothesis): Please formulate the specific non-inferiority hypothesis. The description does not specify whether it should be proven that MCSL is non-inferior to OA-IP or vice versa in relation to the non-inferiority margin of 8%.

Answer: Thank you for your constructive comments, which are very important for readers to have a comprehensive understanding of the background knowledge of this research field and the rationale of our non-inferiority margin choice. 

Because the oblique axis in-plane method (OA-IP) is currently a widely recognized central venous puncture method, it combines the advantages of short-axis and long-axis approaches, thereby increasing the success rate of puncture. Our puncture method is a novel approach that also integrates the advantages of both short-axis and long-axis techniques. Moreover, our method strategically selects puncture points away from anatomical structures such as the external jugular vein and brachial plexus nerves, making it theoretically safer. Therefore, we intend to demonstrate through a non-inferiority study that our method is not inferior to OA-IP in terms of puncture success rate.

Typically, the non-inferiority margin for propotion is set at 10-15%. Following discussions within our professional group, we have opted for a more stringent threshold of 8%. In other words, we consider that if the success rate of our Modified Combined Short and Long-axis (MCSL) method is below 8% compared to OA-IP, it cannot be deemed non-inferior. We have added the rationale for selecting the 8% threshold to our manuscript based on your advice. (Line 55-72).

3. (analysis of primary endpoint variable (PeV): To proof the non-inferiority hypothesis a one-sided 95% confidence interval should be given. This should be calculated for the ITT as well as PP population to confirm rejection of the null hypothesis. (see abstract and result section)

Answer: Thank the reviewer for the comments. When establishing non-inferiority, we used a two-sided 95% confidence interval (CI), which is equivalent to a one-sided 97.25% CI, which has greater accuracy than a one-sided 95% CI (JAMA. 2012 Dec 26;308(24):2594-604. PMID: 23268518. DOI: 10.1001/jama.2012.87802). In our study, despite three patients not completing the assigned treatment and being categorized as puncture failures (surpassing three puncture attempts (n=2) and arterial puncture (n=1)), it is crucial to note that the non-inferiority indicator focuses on the puncture success rate, and data collection was completed for all patients. Consequently, for the statistical analysis of the non-inferiority indicator, we opted for an Intent-to-Treat (ITT) analysis over a Per-Protocol (PP) analysis.

Appreciating your advice, we conducted a PP analysis in the PeV, revealing a rate difference of 7.1% (MCSL (85/92, 92.4%) vs. OA-IP (81/95, 85.3%)) with a 95% CI of -1.8% to 16.1%. Both ITT and PP analyses affirmed the non-inferior conclusion. (Line 191, 192, 200).

4. L28: Please rephrase: ...primary outcome is the event of first .....

Answer: Thank you for your guidance and help, which is very important to increase the accuracy of presentation and improve the quality and readability of the manuscript. We have made changes according to your guidance (Line 28).

5. L34: see analysis of PeV

Answer: Thank you for your comments. After your reminder, we noticed the inaccuracy in our statement and we have revised the error. Thank you again for your rigorous and careful review, which is important to maintaining academic rigor and the readability of the manuscript. (Line 34).

6. L41: As the register is not accessible, please provide a English translated confirmed entry of data.

Answer: Thank you for your suggestions and comments. Line L41 is the project registration information we submitted, including the registration website (Chinese Clinical Trial Registry (ChiCTR)) and registration number (ChiCTR2100046899). We have provided the website address (https://www.chictr.org.cn/) in the method section. Please enter our registration information according to the steps we provide to enter the registration URL (see attachment 3) (Line 80). 

In addition, we have also shared our metadata on the DRYAD public data platform, and we have provided the URL to enter and download the our metadata (see Data Review URL).

7. L70: see formulate of non-inferiority hypothesis

Answer: Thank you for your commens and reminder. We realize that our previous manuscript did not state the details of the hypothesis clearly. We have already stated the details of the hypothesis (Line 71,72).

8. L72: Compare is misleading term here.

Answer: Thank the reviewer for the comments and your suggestion is greatly appreciated. We have replaced compare with more accurate words. Thank you for your professional guidance on the language of our manuscript, which is very important to increase the readability of the manuscript (Line 73). 

9. L76: Evaluator blinded is not clear. Please explain, who assessed the PeV and how blinding is reached.

Answer: Thank you for your review and for asking this meaningful question. We apologize for our negligence in preparing the manuscript and for not attracting readers' attention in particular. We have listed the outcome indicator evaluation in a separate paragraph. We hope that our statement is clear enough. Thank you again for your valuable opinions (Line 177-182). 

10. L80: Use CONSORT for non-inferiority

Answer: Thank the reviewer for the comments. We have added a non-inferiority CONSORT statement, reported our research based on the non-inferiority CONSORT statement, and cited the corresponding literature. (Line 82-84, see S1 Table). 

11. L94: PBR(4) results in a high number of predictable allocations, here at least 47 up to 95 predictable allocations, in particular with implementation by sealed opaque envelops. This causes allocation bias and need to be discussed in the limitation section.

Answer: Thank you for your professional review and valuable advice. Indeed, if the operator knows that the randomization method is block randomization and the block is 4, he can guess the allocation plan. Therefore, when we generate the randomization plan, it is completed by statistical experts, and the randomization method blinds both the operator and the outcome indicator evaluator. And our random allocation plan is also kept by statistical experts. Unblinding was performed only after the experimental and statistical analyzes were completed.

We regret that our previous manuscript did not clearly explain the specific details of randomization generation and allocation concealment. We have added corresponding descriptions, hoping to reduce misleading to readers (Line 101-109).

12. L99. Please explain the meaning and implementation process to blinding of monitors and stat analysts with respect to detection, performance and attrition bias.

Answer: Thank the reviewer for the comments. To prevent selection bias caused by the data cleaning and analysis process, we blinded data monitoring and data analysis.

Before the data analysis was completed, only the enrollment numbers of subjects were recorded, and the group names were replaced with A and B. We have added the details about blind execution in the manuscript. (Line 107-109). 

13. L144ff: Please describe the time and measurement method of the PeV - PVWP. Here in particular the validity of the measurement methods, i.e. objective measurement methods (measurement free of knowledge about allocated treatment), needs to be described.

Answer: Thank you very much for your careful review and professional suggestions. Accurately describing the experimental details can help readers understand the research, and also help subsequent studies to replicate our research. We have added the judgment criteria and methods of the Pev-PVWP, and cited relevant references. As described previously, we performed judgments on ultrasound videotapes by two raters who were not involved in the procedure. (Line 161, 162, 177-182).

14. L171: I could not follow the sample size calculation. My program results in 186/group using the stated numbers. Please provide details of the input data, statistical rule for calculation of the confidence interval as main analysis and software program used for computation.

Answer: Thank you for your comments and questions, which are necessary for scientific rigor. We used PASS software (version 15) to calculate the sample size of the study, using the sample size calculation formula for non-inferiority studies. Please see attachment 2 for the specific calculation process. We have added sample calculation software information in the Methods section (Line 186, 187). 

15. L173: Please justify the non-inferiority margin, e.g. by expected difference of standard to placebo.

Answer: Thank the reviewer for the comments. Our study diverges from a typical drug study as it doesn't involve a placebo. Certainly, we considered the landmark-guided puncture method as a placebo. Unfortunately, through literature review, our comprehensive review did not identify comparative studies on the success rates of the oblique-axis in-plane puncture (OA-IP) method and the landmark-guided jugular vein puncture method. Additionally, our Modified combined short and long axis (MCSL) puncture method is novel, lacking data for comparison with the landmark-guided internal jugular vein puncture method. Based on data from our unpublished preliminary experiments, both methods (OA-IP and MCSL) exhibited a puncture success rate of approximately 90%. Considering common practices in proportion-based research, the non-inferiority margin is typically set at 10-15%. Following group discussions, we opted for a relatively stringent 8%. We have explained in the manuscript the reasons why we chose 8% as the non-inferior margin (Line 71,72). 

16. L177: Please describe the analysis populations.

Answer: Thank the reviewer for the comments. I'm very sorry. There may be discrepancies in page numbers between the manuscript you reviewed and mine. Line 177 is an explanation of statistical analysis in the methods section. We have described the analysis populations in the methods and results section. Moreover, we have presented the population baseline data of the study in Table 2. If there is anything unclear in the presentation, we hope to get guidance from the reviewers. Thank you again. (Line 86-94, 216). 

17. L186: Rephrase according to analysis of PeV.

Answer: Thank the reviewer for the comments and your suggestion is greatly appreciated. We have revised the manuscript according to your advice(Line 221-223). 

18. L193: Skip statistical tests on baseline data, not meaningful in RCT's.

Answer: Thank you for your advice and expertise. Indeed, in large randomized controlled studies, baseline levels are balanced between groups, so we did not perform statistical analysis. However, we present the baseline data of both groups to facilitate data extraction by future researchers conducting systematic reviews. 

19. L196: Please skip data across group = overall data. Not meaningful.

Answer: Thank the reviewer for the comments and suggestion. thanks for your advice. We've removed lengthy narrative (Line 211-213). 

20. L204: Significance test is not meaningful here. Use the one sided 95% confidence interval for argumentation and show how the CI is similar in ITT and PP.

Answer: Thank the reviewer for the comments. When establishing non-inferiority, a one-sided 95% confidence interval is equivalent to a two-sided 90% confidence interval, and a one-sided 97.25% CI is equivalent to a two-sided 95% confidence interval (CI). This study used a 2-sided 95% confidence interval to determine non-inferiority, which has greater accuracy than a two-sided 90% CI (JAMA. 2012 Dec 26;308(24):2594-604. PMID: 23268518. DOI: 10.1001/jama.2012.87802). In our study, despite three patients not completing the assigned treatment and being categorized as puncture failures (surpassing three puncture attempts (n=2) and arterial puncture (n=1)), it was crucial to note that the non-inferiority indicator focuses on the puncture success rate, and data collection was completed for all patients. Consequently, for the statistical analysis of the non-inferiority indicator, we opted for an Intent-to-Treat (ITT) analysis over a Per-Protocol (PP) analysis 

Appreciating your advice, we conducted a PP analysis, revealing a rate difference of 7.1% (MCSL (85/92, 92.4%) vs. OA-IP (81/95, 85.3%)) with a 95% CI of -1.8% to 16.1%. Both ITT and PP analyses affirmed the non-inferior conclusion (Line 191, 192, 200, 245, 246).

21. L273: Avoid reporting of results in the discussion section.

Answer: Thank the reviewer for the comments. We strongly agree with your point that the results should not be reported in the discussion section. So, you don't even see a number in our discussion section.

The description of the regression analysis in the discussion is not a report of the research results; rather, it is our discussion and analysis of these results. Thank you again for your careful review and we greatly admire your hard work. (Line 291-298). 

22. L281: Limitation section should focus on operational and methodological aspects of the study. Include discussion of sample size justification, if needed and allocation bias.

Answer: Thank the reviewer for the comments. We have updated the limitations of our study to state that our study was conducted by only one physician, which is indeed a bias. In the future, we will consider this issue and design a more reasonable study plan. Regarding the issues of sample size and allocation bias, please see the previous answers section for details. If the reviewer has any questions or suggestions, please contact me and give your valuable suggestions so that we can jointly improve the quality of the manuscript. Thank you again. (Line 299-301).

---

## [Decision Letter · Decision Letter 2]

4 Dec 2023

Modified combined short and long axis method versus oblique axis method in adult patients undergoing right internal jugular vein cannulation: A randomised controlled non-inferiority study

PONE-D-23-27163R2

Dear Dr. Tang,

We’re pleased to inform you that your manuscript has been judged scientifically suitable for publication and will be formally accepted for publication once it meets all outstanding technical requirements.

Kind regards,

Luigi La Via

Academic Editor

PLOS ONE

Additional Editor Comments (optional):

Reviewers' comments:

Reviewer's Responses to Questions

**Comments to the Author**

1. If the authors have adequately addressed your comments raised in a previous round of review and you feel that this manuscript is now acceptable for publication, you may indicate that here to bypass the “Comments to the Author” section, enter your conflict of interest statement in the “Confidential to Editor” section, and submit your "Accept" recommendation.

Reviewer #4: All comments have been addressed

2. Is the manuscript technically sound, and do the data support the conclusions?

Reviewer #4: Yes

3. Has the statistical analysis been performed appropriately and rigorously? 

Reviewer #4: Yes

4. Have the authors made all data underlying the findings in their manuscript fully available?

Reviewer #4: Yes

5. Is the manuscript presented in an intelligible fashion and written in standard English?

Reviewer #4: (No Response)

6. Review Comments to the Author

Reviewer #4: The paper is now sound and all comments are adressed. However, substitute "bilateral" by "Two sided" in Line 195.

7. PLOS authors have the option to publish the peer review history of their article (what does this mean?). If published, this will include your full peer review and any attached files.

Reviewer #4: No

---

## [Editor Report · Acceptance letter]

6 Dec 2023

PONE-D-23-27163R2 

Modified combined short and long axis method versus oblique axis method in adult patients undergoing right internal jugular vein cannulation: A randomized controlled non-inferiority study 

Dear Dr. Tang:

I'm pleased to inform you that your manuscript has been deemed suitable for publication in PLOS ONE. Congratulations! Your manuscript is now with our production department. 

Kind regards, 

on behalf of

Dr. Luigi La Via 

Academic Editor

PLOS ONE